# Evaluation of a credit-bearing online administered happiness course on undergraduates' mental well-being during the COVID-19 pandemic

**Catherine Hobbs**[1], **Sarah Jelbert**[1], **Laurie R. Santos**[2], **Bruce Hood**[1]*

**1** School of Psychological Science, University of Bristol, Bristol, United Kingdom, **2** Department of Psychology, Yale University, New Haven, Connecticut, United States of America

* bruce.hood@bristol.ac.uk

**Data Availability Statement:** Data cannot be shared publicly because participants did not consent to data being made open access. Data are available from the University of Bristol Research

## Abstract

Psychoeducational courses focused on positive psychology interventions have been shown to benefit student well-being. However, since the onset of the COVID-19 pandemic and accompanying social restrictions, many educators have had to deliver their courses online. Given that online teaching presents a very different university experience for students, do psychoeducational courses provide similar well-being benefits in an online format? In this pre-registered study (https://osf.io/3f89m), we demonstrate that despite the challenges of remote learning, first year university students (N = 166) taking an online "Science of Happiness" course during the first term experienced positive benefits to mental well-being in comparison to a wait-list control group (N = 198) registered to take the course in the second term. Specifically, university students currently taking the course maintained their mental well-being over the semester relative to the wait-list control who showed a significant decline in well-being and increase in anxiety during the same period. Our findings suggest that the online-administered "Science of Happiness" course delivered during the COVID-19 pandemic was associated with a protective effect on mental well-being. We also observed that engagement with the course was high, though there was no evidence that this factor mediated the positive effects we observed. However, we did find evidence that prior interest in increasing well-being influenced the effects of the course; participants with lower well-being interest showed less of a benefit. Our results suggest that online psychoeducational courses might provide a relatively cheap, flexible, and efficient means of providing support as part of an integrated approach to student mental well-being.

## Introduction

Over the last decade, higher education settings have become more aware of the need to address not only development of academic skills, but also psychological well-being. University students are at high risk for mental health problems, both due to their age range which is considered a

Data Repository for researchers who meet the criteria for access to confidential data (https://doi.org/10.5523/bris.2o4avcpdbwzi02a5fffc35vtql).

**Funding:** This work was supported by a grant awarded jointly by the Elizabeth Blackwell Institute, University of Bristol and the Rosetrees Trust (grant code: R105121-248). The funder had no role in study design, data collection and analysis, decision to publish, or preparation of the manuscript.

**Competing interests:** The course being evaluated was taught by two of the authors (Bruce Hood and Sarah Jelbert), and is based on a similar course developed, and previously taught, by another author (Laurie R. Santos). This does not alter our adherence to PLOS ONE policies on sharing data and materials. Catherine Hobbs has no competing interests.

vulnerable period for mental health, and major life transitions associated with starting university [1–3]. Mental health issues are both highly prevalent in student populations and rising. In a large-scale survey of 37,500 students across 140 universities in the United Kingdom, 22% of students reported a current mental health diagnosis and 88% reported feelings of anxiety [4]. Similarly, a recent 2018 national survey found that more than 40% of U.S. college students report being too depressed to function and more than 10% seriously considered suicide in the last year [5]. In a global survey across eight countries, 31% of students screened positive for a mental health disorder [6].

The critical situation surrounding university student mental health was further exacerbated during the COVID-19 pandemic. Many universities were forced to deliver courses online, dramatically changing students' university experience [7]. Students have had to adapt to new methods of learning, reduced social contact, extended periods of quarantine and increased financial pressures [7, 8]. Recent research shows that these changes have had a substantial impact on students' mental health; one study found that depression and anxiety in university students was at its highest during the pandemic compared to any other time in the academic year [9]. In the UK, 63% of students reported a decline in their mental health in the 2020 autumn academic term [10]. Similar findings of heightened mental distress among higher education students during the pandemic has been reported worldwide [11–16].

Universities typically take a reactive, individualised approach to treating mental well-being; students contact university well-being services when they begin experiencing mental health issues. However, well-being services are increasingly overwhelmed. In 2017 over 90% of universities reported increased demand within the last five years, contributing to long wait-times [17]. Additionally, stigma, and practical difficulties of accessing services often prevent students from seeking help [18, 19]. Despite high rates of mental distress, students are less likely to receive treatment for mental health issues compared to the general population [20]. Additionally, by focusing on students already experiencing issues, current services neglect addressing the development of mental well-being in the wider student population [21].

Faced with these difficulties, some universities have sought an alternative approach, emphasizing preventative community-wide care by developing new academic courses focused on mental well-being. Such courses tend to define mental well-being as a multi-faceted concept, one that encompasses both hedonic and eudemonic well-being. Hedonic well-being relates to the perceived attainment of pleasure, happiness and life satisfaction, and avoidance of pain or negative emotions. In contrast, eudemonic well-being relates to an individuals' experience of meaning, growth, purpose, and authenticity in life [22]. Mental well-being is associated with a range of positive outcomes including higher income [23], greater job satisfaction [24], better health outcomes [25], and lower mortality [26].

Course-based approaches to improving mental well-being ensure that student time is allocated to activities designed to enhance their well-being, increasing engagement and retention compared to voluntary well-being programmes offered outside of the main university curriculum that typically have high attrition [27]. Additionally, an academic course-based approach can reduce stigma associated with mental well-being interventions by having all students participate in the psychoeducational training as part of their degree requirements [28]. From a university perspective, these courses are inexpensive and relatively simple to deliver as they make use of existing administrative infrastructure and can benefit large numbers of students concurrently [27]. Furthermore, research has shown that improving student well-being has wider beneficial effects on student academic performance [29]. Well-being courses embedded within university degree programmes are therefore uniquely placed to benefit student mental well-being and academic performance.

Several university well-being courses have taken a psychoeducational approach focused on Positive Psychology. Students are provided with an evidenced-based understanding of psychological well-being and scientific approaches to increasing happiness [30]. These scientific approaches commonly include positive psychology interventions (PPIs); short activities designed to promote positive emotions and experiences. Systematic reviews have concluded that PPIs are effective in promoting psychological well-being across a range of settings [31]. In particular, PPIs have been widely implemented in schools [32], with evidence of beneficial effects on well-being [33], depression [34], and academic performance [35]. Research on the benefits of PPIs within university degree programmes is a smaller but growing field. A number of studies have reported various benefits of positive psychology courses, including increases in well-being, life satisfaction, happiness and stress management [27, 36–39]. However, other studies have reported inconsistent or null effects [21, 40–42].

This research team previously studied the effects of an 11-week Positive Psychology course, entitled 'The Science of Happiness', on student well-being at the University of Bristol [43]. This course had been modelled on a similar course "Psychology and the Good Life" developed at Yale University by one of the co-authors which had proven to be highly popular. Students who completed the Science of Happiness course at the University of Bristol showed a significant increase in psychological well-being and a decrease in loneliness compared to a wait-list comparison group [43]. These effects were maintained six weeks following the course. Additionally, we conducted an abbreviated 4-week online-version of this course during the COVID-19 pandemic that was open to all staff and students at the University Bristol [43]. We found that this online short-course had a beneficial effect; participants showed an increase in psychological well-being across several measures following completion of the course. However, this study was not pre-registered and did not include a control group. Thus, these previous results must be interpreted with caution.

In the current study, we evaluated the effect of 'The Science of Happiness' course on first-year undergraduate student mental well-being during the COVID-19 pandemic. Within the period of this study (October 2020 –February 2021), COVID-19 regulations included two national lockdowns and regional tier restrictions [44]. Restrictions included social distancing (remaining 2 metres apart from others), reduced social contact (no meeting indoors and restricted numbers of people allowed to meet outdoors), and closures of hospitality venues. Students had the option to reside in university accommodation or at home and had access to some university campus resources. Prior to COVID-19, teaching at further and higher education institutes was predominantly delivered face-to-face. However due to COVID-19 restrictions, The Science of Happiness course evaluated in this study, alongside most university courses, was taught entirely online during this period.

Previous work has shown that psychoeducational courses delivered as part of massive open online courses can provide a significant benefit to learners' mental health [45], but to the best of our knowledge the present study is the first to test whether similar online content can improve well-being in a university academic course setting. Our current intervention built on previous research by delivering the full 11-week programme remotely and including a wait-list comparison group to ensure observed effects were not due to natural temporal well-being fluctuations unrelated to the course content. As previous research has reported that well-being courses may improve academic performance [35], we also evaluated the impact of the course on students' perceptions of academic performance. As stated in our pre-registration, we hypothesised that completion of the course would have a positive effect on student mental well-being and perceptions of academic performance, compared to a wait-list control group.

## Materials and method

This study was pre-registered on Open Science Framework (https://osf.io/3f89m). The data that support the findings of this study are available in the University of Bristol Research Data Repository upon request (https://doi.org/10.5523/bris.2o4avcpdbwzi02a5fffc35vtql). Code for statistical analyses is openly available on Open Science Framework (https://osf.io/w25nm).

### Ethics

This research was approved by the University of Bristol School of Psychological Sciences Research Ethics Committee (Approval code: 011020110763). Participants provided informed written consent as part of an online survey during data collection.

### Participants

Participants were first year undergraduate students at the University of Bristol, United Kingdom that had chosen to take the credit-bearing course, 'The Science of Happiness', in the academic year 2020/21. Participants were required to be aged 18 and over and could not proceed with the survey without confirming their age.

The intervention group consisted of students allocated to complete the course in the first semester of the academic year (October-December 2020). The wait-list control group consisted of students allocated to complete the course in the second semester of the academic year (February-May 2021). Assignment was not randomised due to restrictions with academic timetabling but was conducted independently of the research team by administrative staff. Students were assigned to complete the course in the first or second semester dependent on time-tabling restrictions within degree programmes. At the point of enrolment students were not aware of which semester they would be assigned to complete the course.

All students enrolled in the course were invited to participate in this research. Participation was optional and students were still eligible to complete 'The Science of Happiness' course if they did not choose to participate in this study. A priori sample size calculations indicated that we required 94 participants per group to detect effects of a similar magnitude to those previously observed ($\eta p2$ = .041) [43] with 80% power at an alpha level of 0.05.

### Procedure

**'The Science of Happiness' course.**   Participants in the intervention group completed the 11-week course 'The Science of Happiness'. The course was similar in format and content as was delivered in the previous academic year [43]. However, due to the COVID-19 pandemic and accompanying social restrictions, the course was delivered entirely online.

Students were asked to watch 10 pre-recorded lectures on a weekly basis. Lectures were uploaded each week but could be viewed at any time by students once available. The lecture series was led by one of the study authors (BH), a professor of psychology. The series began by introducing students to the concepts of happiness and well-being, before examining different theories of promoting mental well-being. Topics covered included signature strengths, meditation, kindness, gratitude, exercise, sleep, social connections, and social comparisons. An overview of psychological research methods was also provided to allow students to evaluate the evidence base underlying these approaches.

To encourage social contact that would typically be encountered during in-person lectures, students attended 11 weekly live sessions delivered via online video conferencing. One live session was scheduled per week with all students attending the same session simultaneously. The live sessions were led by two of the study authors (SJ and BH), a lecturer and professor of

psychology, respectively. The live session revisited the week's pre-recorded lecture content, elaborating on examples and including activities such as pop quizzes, Q&As, replicating simple online experiments and group discussions. Individuals' attendance at live sessions was not monitored, but overall attendance was typically around 80%. Recordings of the live session were also made available to students.

Students were also required to participate in weekly 'happiness hubs', led by senior students and post-graduate mentors. Each group consisted of a maximum of eight students who met for approximately one hour per week using online video conferencing. In these 'happiness hubs', students were encouraged to implement the scientifically-validated PPIs taught in lectures and live sessions.

Finally, students completed online journal entries relating to PPIs that they were asked to implement in their own time each week. These included three good things [46], learned optimism [47], a gratitude letter [48], acts of kindness [49], signature strengths [50] and goal setting [51].

Students earned course credit through participation in the course with a minimum level of attendance at happiness hubs, viewing of pre-recorded lectures, and completion of journal entries required.

**Data collection.** Participants completed self-report measures using the online survey software Qualtrics [52] at three timepoints: (time 1) the beginning of semester 1, October 2020; (time 2) the end of semester 1, December 2020; and (time 3) the beginning of semester 2, February 2021. Links to complete the surveys were circulated to students using announcements on the University of Bristol online learning environment within the Science of Happiness course pages for both groups. Reminder announcements were circulated approximately one week after the initial invitation for each timepoint. Responses were required within two weeks of the surveys being opened.

For the intervention group these timepoints corresponded to pre-completion of the course, post-completion of the course, and a six week-follow up. For the wait-list control group all timepoints occurred prior to completion of the course.

## Measures

**Well-being.** We used the Short Warwick-Edinburgh Mental Well-Being Scale (SWEMWBS) as a measure of mental well-being at each timepoint [53]. The SWEMWBS uses 7-items of the Warwick-Edinburgh Mental Well-Being Scale (WEMWBS). We used metric scores converted following standard procedures [54], to allow comparison to the full version of the WEMWBS. Scores range from 7 to 35, with greater scores indicating greater mental well-being. The SWEMWBS has been validated for use in the general population [55].

The Subjective Happiness Scale (SHS) was used to measure happiness at each timepoint [56]. The SHS is a 4-item scale using a 7-point Likert Scale, with responses averaged across items. Scores range from 1 to 7. Greater scores indicate greater levels of happiness. The SHS shows good levels of reliability and validity [56].

We also used the Generalised Anxiety Disorder Questionnaire (GAD-7) as a measure of anxiety at each timepoint [57]. The GAD-7 consists of 7-items relating to experience of generalised anxiety symptoms in the previous two weeks. Scores range from 0 to 21, with greater scores indicating greater anxiety. The psychometric properties of the GAD-7 have been established in the general population [58].

At timepoints 2 and 3, we used a single-item Global Rating of Change (GRC) scale, to measure change in participants' perceptions of mental well-being. Participants were asked *'Compared to the [start of term/the academic year], how has your mental well-being changed*?'. Possible responses ranged from (1) 'I feel a lot better' to (5) 'I feel a lot worse'. GRC scales are

widely used in clinical practice and research to assess patients' perceptions of change in well-being or health [59].

Participants in the intervention group also provided a weekly measure of happiness when submitting their journal entries, derived from the Office of National Statistics' subjective well-being measure [60]. Participants rated how happy they felt on the previous day on a scale from 0 ("Not at all") to 10 ("Completely").

**Perceptions of academic performance.** We measured perceptions of academic performance at timepoints 2 and 3, through two self-report items ('*How [satisfied/worried] are you with your academic performance at the University to date*?'), using a five-point Likert scale ranging from (1) 'Not at all' to (5) 'A great deal'.

**Positive expectations and interest.** At timepoint 1, we asked participants to what extent they agreed with the statement '*I feel positive about being at University*', as a measure of positive expectations of university, and '*I am actively interested in trying activities that could increase my well*-being' as a measure of prior interest in well-being. Possible responses ranged from (1) 'Strongly Agree' to (7) 'Strongly Disagree'.

**Engagement.** To determine student engagement, we recorded the number of journal entries completed (maximum of 10), pre-recorded lectures viewed (maximum of 20) and happiness hubs attended (maximum of 11).

**Feedback.** Following completion of the course at timepoint 2, we asked the intervention group whether they would recommend the course to other students (yes or no). We also asked participants to what extent they agreed that the course was '*interesting and informative*' and '*enjoyable*', and to what extent they felt that the course had a '*positive effect on [their] mental well-being*'. Possible responses ranged from (1) 'Strongly Agree' to (5) 'Strongly Disagree'.

**Impact of COVID-19 pandemic.** To contextualise our findings within the COVID-19 pandemic we asked participants to what extent they agreed with the statement '*I have been experiencing excess stress or anxiety due to the impact of COVID-19*'. Possible responses ranged from (1) 'Strongly Agree' to (5) 'Strongly Disagree'.

## Statistical analyses

R version 4.0.3 was used for all analyses.

### Hypothesis 1: Participation in the online Science of Happiness course will have a positive effect on student mental well-being, compared to students that have not yet taken this course.

We used mixed effects linear regression models to investigate the influence of the Science of Happiness course on student mental well-being. Group, timepoint and an interaction term between group and timepoint were used as predictors. Participant was entered as a random effect to account for the within-subject measurement of time. SWEMWBS, SHS, and GAD-7 scores, were entered as continuous outcomes in individual models. Tukey adjusted follow-up pairwise comparisons were used to investigate changes over time according to group where significant interaction effects were observed.

### Hypothesis 2: Participation in the online Science of Happiness course will have a positive effect on students' perceptions of overall academic performance, compared to students that have not yet taken this course.

We used mixed-effect linear regression models with perceptions of academic performance (worry, satisfaction) as continuous outcomes in individual models. Group (intervention, waitlist) was entered as a predictor. To account for data being collected at multiple timepoints, timepoint (2, 3) was entered as a fixed effect, and subject was entered as a random effect.

**Does the influence of the Science of Happiness course on student mental well-being differ according to method of delivery (in-person vs. online)?.**   We also examined whether the influence of the Science of Happiness on student mental well-being during the academic year 2020/21 (the focus of this study) where the course was delivered online during the COVID-19 pandemic, differed from the previous academic year (2019/20) where the course was delivered in-person prior to the COVID-19 pandemic [43]. SWEMWBS scores were used as a continuous outcome measure, and timepoint (1, 2, 3), group (intervention, wait-list control) and year/method of delivery (2019/20 in-person, 2020/21 online), and interaction terms between these variables were entered as predictors. Participant was entered as a random effect to account for within-subject measures.

**Exploratory analyses.**   Due to an unanticipated imbalance between groups in degree programme, we repeated the analyses for hypotheses one and two adjusting for degree programme by including a binary variable indicating whether participants were enrolled on a Psychology degree or another degree as an additional predictor in the regression models.

We used mixed-effects linear regression models to examine change in weekly happiness ratings provided by the intervention group. As journal entries were only mandatory from week two, a limited number of participants provided data at week one (n = 58). To reduce potential self-selection bias we only included ratings from week two onwards in our analyses. We entered happiness ratings as a continuous outcome, week as a continuous predictor, and subject as a random-effect to account for the repeated-measure nature of our data.

We also used an anchor-based method with the GRC applied to SWEMWBS scores to quantify the minimal important change in mental well-being to aid power calculations for future research [61]. Mean and standard deviations of change and standardised mean differences (Cohen's dz) in SWEMWBS scores from timepoints 1 to 2 and 1 to 3 were calculated according to responses on the GRC. We conducted a sample size calculation based on 80% power and an alpha level of 0.05 to determine the minimum sample required to detect mean changes in SWEMWBS scores in participants' who reported feeling 'slightly better'.

Finally, we examined whether level of engagement in the course, positive expectations of university and interest in increasing well-being influenced the effects of the Science of Happiness course on well-being in the intervention group. To create a composite measure of engagement we computed z-scores of our individual measures of engagement and summed the values. Possible values of this engagement index varied between 0 to 3, with greater engagement in the course indicated by higher values. We used mixed-effect linear regression models with changes in self-report measures of well-being (SWEMWBS, GAD-7 and SHS scores) as continuous outcomes in separate models. We entered the engagement index, positive expectations of university and interest in increasing well-being as continuous predictors in individual models. As change in well-being is likely to be dependent on baseline level of well-being [27], we adjusted for timepoint 1 scores. To account for data being collected at multiple timepoints, timepoint (2, 3) was entered as a fixed effect, and subject as a random effect.

## Missing data

We handled missing data in our mixed-effect linear regression models using maximum likelihood estimation. For our estimations of the minimal important change in well-being we used complete case analysis.

## Results

### Sample characteristics

At timepoint 1, 176 participants in the intervention group and 208 participants in the wait-list control group provided data. Of those in the intervention group, 145 (82%) provided data at

timepoint 2 and 147 (83.5%) provided data at timepoint 3. Of those, in the wait-list control group, 154 (74.0%) provided data at timepoint 2 and 191 (91.8%) provided data at timepoint 3. We included participants in our main analyses that provided data at timepoint one and at least one follow-up timepoint, resulting in a sample of 166 (94.3%) participants in the intervention group and 198 (95.2%) in the wait-list control group.

Sample characteristics are outlined in Table 1. Groups were similar in age, gender, ethnicity, and nationality. However, due to timetabling constraints groups were unbalanced in relation to degree programme. The majority of participants in the intervention group were enrolled in a psychology degree (69.28%), compared to a minority of those in the wait-list group (24.75%).

### Hypothesis 1: Participation in the online Science of Happiness course will have a positive effect on student mental well-being, compared to students that have not yet taken this course.

Descriptive statistics for measures of student mental well-being are available in Table 2.

**Mental well-being.** Change in SWEMWBS scores between timepoints 1 and 2 significantly differed according to group ($\beta$ = -1.25, 95% CI: -1.90, -0.60, p < .001). This effect was maintained at timepoint 3 ($\beta$ = -0.87, 95% CI: -1.50, -0.25, p = .007; Fig 1). Follow-up pairwise comparisons indicated no significant change in SWEMWBS scores over time for the intervention group (timepoint 1 to 2: $\beta$ = 0.26, 95% CI: 0.31, 0.82, p = 0.536; timepoint 1 to 3: $\beta$ = -0.31, 95% CI: -0.88, 0.25, p = 0.390). However, in the wait-list control group SWEWMBS scores decreased on average by 1.00 points by timepoint 2 (95% CI: - 1.54, - 0.45, p < .001), and 1.19 points by timepoint 3 (95% CI: -1.69, -0.69, p < .001).

**Table 1. Sample characteristics according to group.**

| | Intervention (n = 166) | Wait-List (n = 198) |
|---|---|---|
| Age, M (SD)[a] | 18.91 (2.25) | 18.86 (1.30) |
| Gender, N (col%) | | |
| Female | 127 (76.51) | 168 (84.85) |
| Male | 36 (21.69) | 28 (14.14) |
| Non-Binary | 3 (1.81) | 0 (0) |
| Undisclosed | 0 (0) | 2 (1.01) |
| Ethnicity, N (col%) | | |
| White | 129 (77.71) | 154 (77.78) |
| Asian | 22 (13.25) | 17 (8.59) |
| Black | 7 (4.22) | 9 (4.55) |
| Mixed | 3 (1.81) | 8 (4.04) |
| Arab | 3 (1.81) | 1 (0.51) |
| Other | 1 (0.60) | 4 (2.02) |
| Undisclosed | 1 (0.60) | 5 (2.53) |
| Nationality, N (col%) | | |
| British | 128 (77.11) | 157 (79.29) |
| Other | 35 (21.08) | 37 (18.69) |
| Undisclosed | 3 (1.81) | 4 (2.02) |
| Degree Programme, N (col%) | | |
| Psychology | 115 (69.28) | 49 (24.75) |
| Other | 47 (28.31) | 147 (74.24) |
| Undisclosed | 4 (2.41) | 2 (1.01) |

[a] Age was not disclosed by one participant in the intervention group and two participants in the wait-list group.

**Table 2. Mental well-being, anxiety and happiness according to timepoint and group.**

| | | Time 1 | | | | | | Time 2 | | | | | | Time 3 | | | | | |
| | | Intervention | | | Wait-List | | | Intervention | | | Wait-List | | | Intervention | | | Wait-List | | |
| | | M | SD | N | M | SD | N | M | SD | N | M | SD | N | M | SD | N | M | SD | N |
|---|---|---|---|---|---|---|---|---|---|---|---|---|---|---|---|---|---|---|---|
| Online 2020/21 | SWEMWBS | 21.34 | 3.09 | 166 | 21.53 | 3.35 | 198 | 21.63 | 2.88 | 145 | 20.44 | 3.38 | 154 | 21.02 | 3.19 | 147 | 20.33 | 3.24 | 191 |
| | GAD-7 | 7.46 | 4.55 | 166 | 7.58 | 4.91 | 196 | 7.61 | 4.61 | 145 | 9.03 | 5.19 | 153 | 7.34 | 4.73 | 145 | 8.77 | 5.28 | 189 |
| | SHS | 4.34 | 1.05 | 166 | 4.34 | 1.17 | 198 | 4.44 | 1.06 | 145 | 4.17 | 1.15 | 153 | 4.31 | 1.07 | 146 | 4.18 | 1.17 | 190 |
| In-Person 2019/20 | SWEMWBS | 20.91 | 2.70 | 135 | 21.36 | 2.96 | 137 | 22.42 | 3.13 | 130 | 21.33 | 3.41 | 177 | 22.10 | 3.24 | 123 | 21.45 | 3.03 | 242 |

M = Mean, SD = Standard Deviation, N = Number of participants.

**Anxiety.** We found evidence of an interaction between timepoint and group on GAD-7 scores at both timepoints 1 to 2 ($\beta$ = 1.11, 95% CI: 0.14, 2.08, p = 0.025) and timepoints 1 to 3 ($\beta$ = 1.27, 95% CI: 0.33, 2.20, p = 0.008; Fig 2). Follow-up pairwise comparisons indicated that participants in the intervention group showed no change in GAD-7 scores (timepoint 1 to 2: $\beta$ = 0.08, 95% CI: -0.75, 0.92, p = 0.970, timepoint 1 to 3: $\beta$ = -0.11, 95% CI: -0.94, 0.73, p = 0.951), whereas those in the wait-list control group showed an increase in GAD-7 scores over time (timepoint 1 to 2: $\beta$ = 1.20, 95% CI: 0.39, 2.00, p = 0.002, timepoint 1 to 3: $\beta$ = 1.16, 95% CI: 0.42, 1.90, p = 0.001).

**Happiness.** We found some evidence that change in SHS scores between timepoints 1 and 2 differed according to group ($\beta$ = -0.17, 95% CI: -0.33, -0.01, p = 0.033; Fig 3). However, follow-up comparisons indicated no evidence of change over time in either the intervention group ($\beta$ = 0.06, 95% CI: -0.08, -0.20, p = 0.539) or wait-list group ($\beta$ = - 0.11, 95% CI: -0.024, 0.02, p = 0.118). We found no evidence of an interaction effect of group with timepoints 1 to 3 ($\beta$ = -0.12, 95% CI: -0.27, 0.04, p = 0.137).

**Hypothesis 2: Participation in the online Science of Happiness course will have a positive effect on students' perceptions of academic performance, compared to students that have not yet taken this course.**

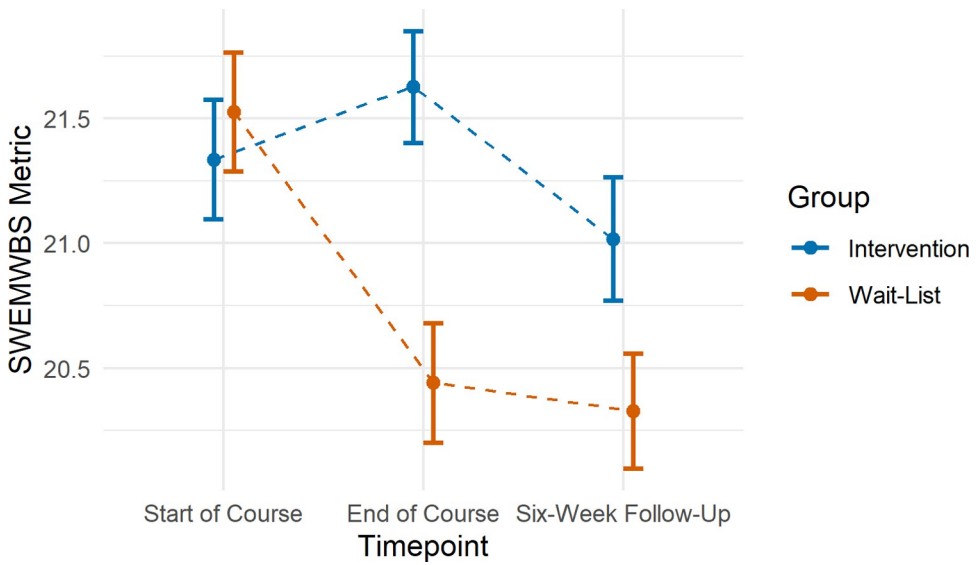

**Fig 1. Mean SWEMWBS scores according to timepoint in the intervention and wait-list comparison groups.** Error bars represent standard errors.

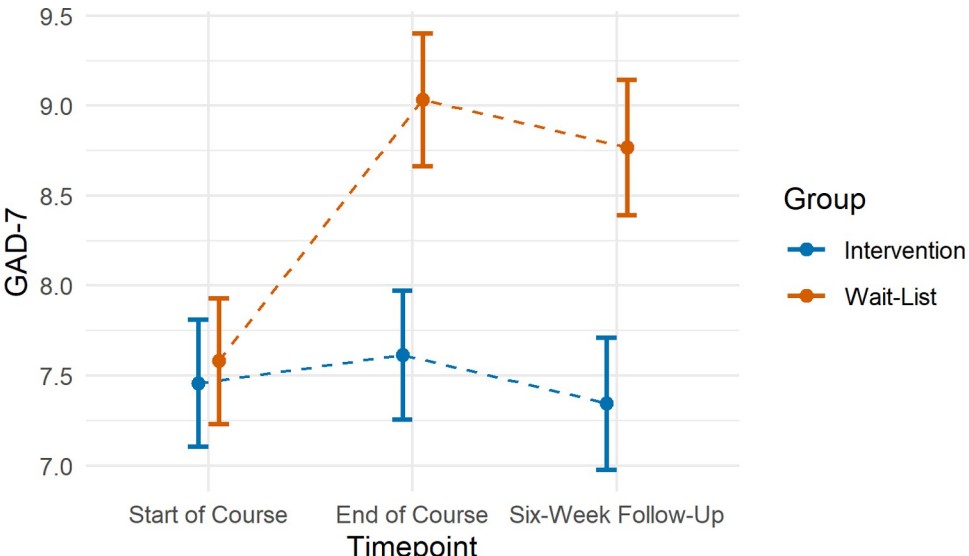

**Fig 2. Mean GAD-7 scores according to timepoint in the intervention and wait-list comparison groups.** Error bars represent standard errors.

We did not find evidence to support our hypotheses; satisfaction with, or worry about, academic performance at follow-up timepoints did not vary by group (satisfaction: β = 0.78, 95% CI: -0.19, 0.14, p = 0.779; worry: β = 0.10, 95% CI: -0.11, 0.30, p = 0.370). Descriptive statistics are reported in Table 3.

## Adjustment for degree programme

To investigate whether our results may be attributable to a greater proportion of participants in the intervention group completing a psychology degree, we repeated our analyses for hypotheses one and two, adjusting for degree programme. The effects previously described

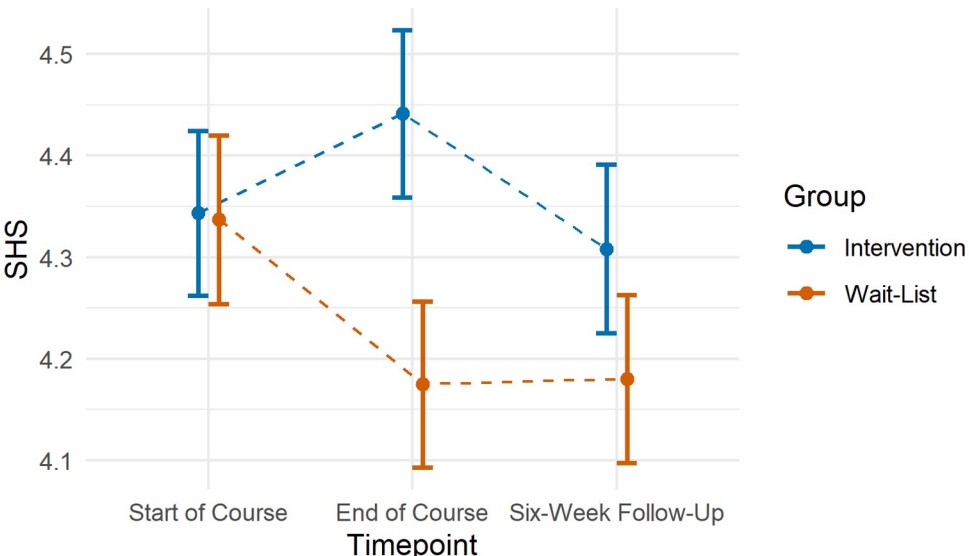

**Fig 3. Mean SHS scores according to timepoint in the intervention and wait-list comparison groups.** Error bars represent standard errors.

**Table 3. Satisfaction and worry about academic performance according to timepoint and group.**

| | Timepoint 2 | | Timepoint 3 | |
|---|---|---|---|---|
| | Intervention | Wait-List | Intervention | Wait-List |
| | (n = 144) | (n = 153) | (n = 143) | (n = 188) |
| Satisfaction, N (%) | | | | |
| Not at all | 12 (8.33) | 17 (11.11) | 6 (4.20) | 18 (9.57) |
| A little | 40 (27.78) | 42 (27.45) | 40 (27.97) | 38 (20.21) |
| A moderate amount | 75 (52.08) | 67 (43.79) | 67 (46.85) | 94 (50.00) |
| A lot | 16 (11.11) | 24 (15.69) | 26 (18.18) | 33 (17.55) |
| A great deal | 1 (0.69) | 3 (1.96) | 4 (2.80) | 5 (2.66) |
| Worry, N (%) | | | | |
| Not at all | 10 (6.94) | 8 (5.23) | 7 (4.90) | 15 (7.98) |
| A little | 49 (34.03) | 51 (33.33) | 52 (36.36) | 57 (30.32) |
| A moderate amount | 48 (33.33) | 37 (24.18) | 39 (27.27) | 63 (33.51) |
| A lot | 25 (17.36) | 37 (24.18) | 33 (23.08) | 37 (19.68) |
| A great deal | 12 (8.33) | 20 (13.07) | 12 (8.39) | 16 (8.51) |

were maintained. Furthermore, degree programme was not associated with self-report measures of well-being or perceptions of academic performance. Full results are provided in S1 File.

### Did weekly happiness ratings increase during the course?

In the intervention group, 147 (88.55%) participants provided one or more weekly happiness ratings. On average, happiness ratings significantly increased throughout the course (Fig 4; β = 0.08, 95% CI: 0.05, 0.11, p < .001).

### What is the smallest change in mental well-being considered meaningful to participants?

The smallest effect size of interest is commonly determined by identifying change in outcomes of interest in relation to participants' perceptions of feeling 'a little better' [61]. Participants in

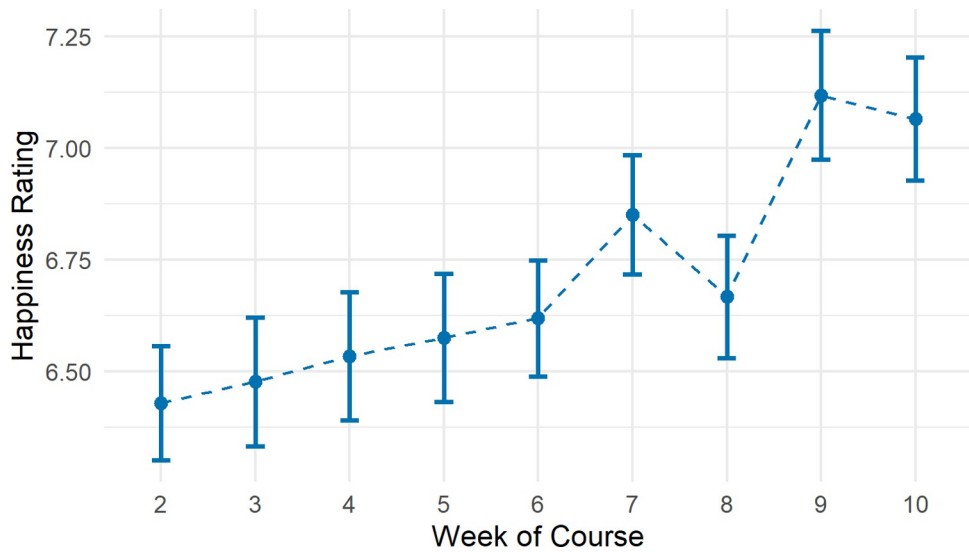

**Fig 4. Mean happiness ratings by week in the intervention group.** Error bars represent standard errors.

the intervention group who reported feeling 'a little better' on the GRC after completing the Science of Happiness course showed a mean change in SWEMWBS scores of 0.83 points (95% CI: 0.21, 1.44), an effect of Cohen's $d_z$ = 0.35 (95% CI: 0.09, 0.61). A minimum of 67 participants would be required to detect this change, with 80% power at an alpha level of 0.05.

Mean changes according to all GRC responses and timepoints are available in S1 Table.

### Did level of engagement, positive perceptions of university life and interest in increasing well-being influence the effect of the Science of Happiness course on student well-being?

Engagement in the course was high. On average, participants attended 90.85% (SD 10.90%) of happiness hubs, completed 86.20% of journals (SD 11.89%) and viewed 76.57% (SD 25.69%) of pre-recorded lectures. The majority of students agreed or strongly agreed that they felt positive about being at University (N = 107; 64.45%), and that they were interested in activities that could increase well-being (N = 110; 66.26%; full results available in S2 Table).

We did not find evidence that level of engagement in the course influenced change in well-being (SWEMWBS: β = 0.38, 95% CI: -0.42, 1.17, p = 0.353; GAD-7: β = -0.63, 95% CI: - 1.78, 0.53, p = 0.289; SHS: β = 0.17, 95% CI: -0.05, 0.40, p = 0.126). Additionally, positive perceptions of being at university did not influence change in well-being (SWEMWBS: β = -0.21, 95% CI: -0.60, 0.18, p = 0.292; GAD-7: β = 0.21, 95% CI: -0.33, 0.74, p = 0.453; SHS: β = 0.04, 95% CI: -0.06, 0.15, p = 0.416).

However, we found some evidence that prior interest in increasing well-being influenced change. Participants in the intervention group who indicated less interest, showed greater declines in SWEMWBS scores (β = -0.54, 95% CI: -0.89, -0.19, p = 0.003), and greater increases in GAD-7 scores (although confidence intervals came close to overlapping the null; β = 0.51, 95% CI: 0.01, 1.02, p = 0.049). There was no evidence of a relationship between prior interest and change in SHS scores (β = -0.07, 95% CI: -0.18, 0.03, p = 0.170).

### How did participants feel about the 'Science of Happiness' course?

Of participants in the intervention group who provided optional feedback following completion of the course (n = 143; 98.62%), the vast majority indicated that they would recommend the Science of Happiness course to other students (n = 137; 95.80%). Additionally, 89.51% (n = 128) agreed or strongly agreed that they found the course 'interesting and informative', 89.51% (n = 128) agreed or strongly agreed that the course was 'enjoyable', and 60.14% (n = 86) agreed or strongly agreed that the course had a 'positive effect on [their] mental well-being'.

### Does the influence of the Science of Happiness undergraduate course on student mental well-being differ according to method of delivery (in-person vs. online)?

The Science of Happiness course described in this paper took place within the COVID-19 pandemic. The majority of participants agreed or strongly agreed (71.70%) that they had been 'experiencing excess stress or anxiety due to the impact of COVID-19'. To examine whether the Science of Happiness course had a differential impact when taught online during the pandemic (academic year 2020/21, the main focus of this paper), we compared our results to those obtained when the Science of Happiness course was delivered in-person prior to the pandemic (academic year 2019/20) [43].

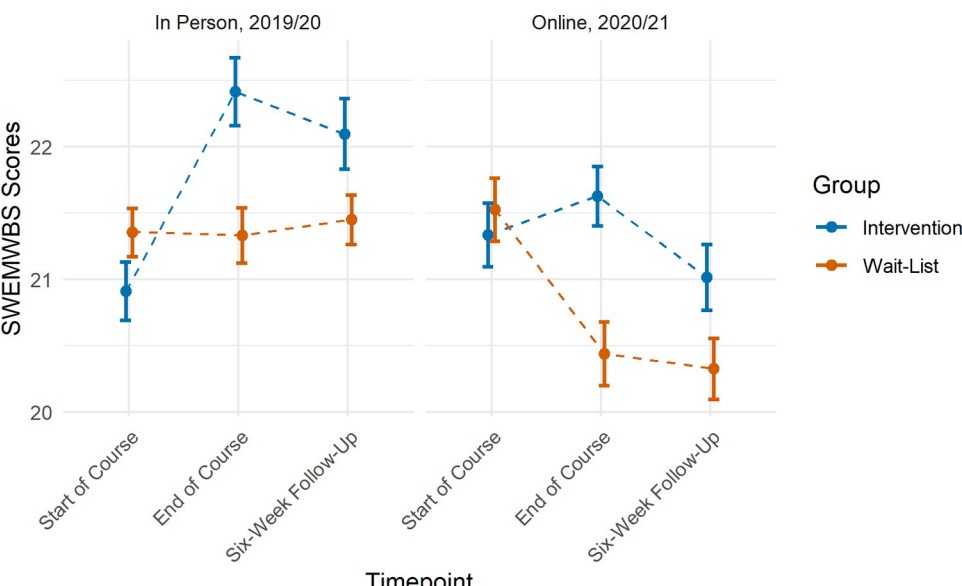

**Fig 5. Mean SWEMWBS scores by timepoint and group.** Results from the academic year 2019/20 where the course was delivered in-person prior to the pandemic are presented in the left panel, and results from the academic year 2020/21 where the course was delivered online during the pandemic are presented in the right panel. Error bars represent standard errors.

We did not find evidence of a three-way interaction between group, timepoint and year/method of delivery (timepoint 1 to 2: β = 0.15, 95% CI: -0.96, 1.27, p = 0.789, timepoint 1 to 3: β = 0.16, 95% CI: -0.93, 1.25, p = 0.776). This suggests that the relationship between SWEMWBS scores and group, as described above and for the previous academic year [43], did not vary by year/method of delivery.

However, we did find evidence of an interaction between timepoint and year/method of delivery (timepoint 1 to 2: β = -1.18, 95% CI: -1.20, -0.37, p = 0.004, timepoint 1 to 3: -1.51, 95% CI: -2.33, -0.69, p < .001). SWEMWBS scores in the current year, in which the course was taught online during the COVID-19 pandemic, on average declined over time. Whereas scores in the previous year, in which the course was taught in-person prior to the COVID-19 pandemic, on average showed an increase (Fig 4).

We also found evidence of an interaction between group and timepoint (timepoint 1 to 2: β = -1.46, 95% CI: -2.28, -0.64, p < .001, timepoint 1 to 3: β = -1.00, 95% CI: -1.81, -0.19, p = 0.015). SWEMWBS scores were on average higher in the intervention group at follow-up timepoints compared to the wait-list group across both in-person and online methods of delivery (Fig 5).

## Discussion

We found evidence that an online credit-bearing psychoeducational course focused on positive psychology interventions that was delivered during the COVID-19 pandemic was associated with a protective effect on students' psychological well-being. Whereas participants in the wait-list control group showed a decline in mental well-being and an increase in anxiety, participants in the intervention groups' mental well-being and anxiety were maintained throughout the course and at 6-week follow-up.

This course was delivered entirely online during the COVID-19 pandemic, a period of increased mental strain. In the United Kingdom, 63% of students reported a decline in their

mental health during the time period of this study [10]. In our study, 71.7% of the overall sample reported that they had experienced excess stress or anxiety due to the pandemic. We explored whether the effects of the Science of Happiness course differed when the course was delivered online during the pandemic compared to the previous academic year, where it was delivered in-person prior to the pandemic [43]. Whilst we observed overall declines in mental well-being in the year of the pandemic (2020/21) compared to the year prior (2019/20), SWEMWBS scores were on average higher in the intervention groups than the wait-list groups at follow-up timepoints across both years. In combination, our findings suggest that the Science of Happiness is associated with beneficial effects on well-being irrespective of method of delivery and is effective even during the time of a worldwide mental health crisis.

However, it is important to note that the positive effects we observed are relatively small in magnitude, at least when considered at a participant-level. On average, the wait-list control group showed a decrease in SWEMWBS and an increase in GAD-7 scores of approximately one point. It is possible that individuals may not recognize a change of this magnitude as psychologically detectable. Indeed, students in the wait-list group who reported 'feeling slightly worse' on average showed a decline in SWEMWBS scores of approximately two points (S1 Table), double the magnitude of the average change we observed across the group. However, well-being courses embedded into university curriculum are intended to be relatively low-intensity interventions, using minimal resources to impact large numbers of students. Additionally, these courses are designed for preventative effects on declines in mental health rather than to treat mental health problems. It is thus expected that such interventions would have a smaller effect on well-being than would be expected with higher-intensity individualised interventions in clinical populations.

In the current study, we employed two measures of happiness that provided contrasting findings; the SHS which participants completed at each timepoint, and a single-item measure of happiness answered weekly throughout the course by participants in the intervention group. In contrast to our expectations, we did not find evidence of change in happiness, measured by the SHS, in either the intervention or wait-list control group. These findings contrast with the exploratory results for our single-item measure, which indicated small increases in happiness in the intervention group. One possibility is that the SHS captures trait levels of happiness that are less susceptible to change. Whereas our single-item measure of happiness asked participants to reflect on the previous day, the SHS asks participants what 'is most appropriate in describing' themselves. Future studies may choose to include a timeframe in the instructions of the SHS to capture state-levels of happiness. It is also possible that the Science of Happiness course was effective at targeting well-being, but not happiness. Whilst these concepts are related, well-being is considered to reflect holistic measure of psychological wellness only part of which includes happiness [62]. Further research is required to clarify the effect of the Science of Happiness course on happiness per se.

Additionally, we did not find evidence to support our hypothesis that the Science of Happiness course would have a positive effect on students' perceptions of academic performance. Participants who completed the course showed comparable levels of satisfaction with, and worry about, academic performance to participants in the wait-list comparison group. These results contrast with previous research documenting positive effects of evidence-based interventions to promoting well-being on academic achievement [35]. However, it is notable that we still observed beneficial effects on mental well-being even in the absence of change in students' perceptions of their own academic performance, given that concerns over academic achievements are one of the strongest drivers of mental distress [18, 63]. The Science of Happiness course may have therefore had a protective effect not only from the mental health implications of the COVID-19 pandemic, but also from stressors arising from academic performance.

Finally, we also explored factors that may have moderated the effects of the course on changes in psychological well-being. Baseline interest in increasing well-being was found to influence change in mental well-being and to some extent anxiety; participants that indicated less interest were less likely to benefit from the course on these measures. However, we did not find evidence that level of engagement in various aspects of the course impacted change in well-being. Our results are largely consistent with previous research [27]. These findings raise the question of whether it is most beneficial to deliver these courses as mandatory or optional courses. Whilst making well-being courses mandatory targets the largest numbers of students and addresses potential stigma, our findings suggest that students with less interest in increasing their well-being are less likely to benefit. Optional courses may be most beneficial in focusing on students that have an active interest in increasing their well-being. Supporting this theory, a recent study that required students to complete a university-wide compulsory well-being course did not report beneficial effects on standardised measures of well-being [40].

## Future research

The Science of Happiness course evaluated within this study is an eleven-week academic course that incorporates multiple PPIs and uses a variety of teaching methods–both active and didactic–including lectures, small group meetings and journaling (see [64] for further discussion of the journaling component in this course). Future research clarifying the active ingredients of the intervention would be worthwhile in order to identify which aspects of the course are most beneficial for students. As there is currently no consistent definition or required elements for educational courses to be considered as mental well-being interventions this may also help inform an evidence-based approach within this field. By identifying which aspects of the course are most important, educators could potentially develop shorter versions of such courses that may still have comparable positive effects on student mental well-being. We have previously reported that an abbreviated four-week version of this course administered during the first COVID-19 national lockdown was associated with increased mental well-being [43]. Whilst this study lacked a control group, limiting conclusions as to the causality of this effect, it is suggestive that smaller-scale interventions may be of benefit. Further research comparing the benefits of a full versus abbreviated version of the course with control groups would be helpful.

## Implications

The findings of this study, in combination with a growing body of evidence [27, 36–39], indicate that psychoeducational positive psychology courses embedded within university curriculum are an effective and efficient method of targeting well-being in students. In this study we have demonstrated that a credit-bearing online Science of Happiness course had a protective effect on student mental well-being during a period of increased mental health issues and stress. Our previous research, conducted prior to the pandemic, documented similar increases in well-being in students that completed an in-person version of the Science of Happiness course [43]. Our work suggests that universities should consider integrating psychoeducational courses focused on positive psychology into curriculum to tackle growing mental health issues and promote positive well-being in students.

To aid future research in this field we additionally estimated the minimal change in well-being perceived by students as important in order to determine the smallest effect size of interest [61]. As with the majority of research on health interventions, previous studies in this field have focused on statistically significant change in well-being. However, whilst small changes may be considered statistically significant, participants may not perceive these changes as

meaningful [65]. Based on the average change in well-being for students that reported feeling slightly better, we recommend future studies include a minimum of 67 participants in intervention groups. This will help to ensure observed changes in well-being are not only statistically significant, but also considered meaningful to students.

## Limitations

Due to timetabling constraints of students electing to take the Science of Happiness course we were unable to randomise participants to groups. It is therefore possible that our results may be attributable to factors other than the Science of Happiness course. However, with the exception of degree programme, the groups had comparable characteristics and baseline levels of well-being. Additionally, when we accounted for our intervention group having a higher proportion of students enrolled in a psychology degree our results were consistent.

Furthermore, our sample was limited in its diversity. Whilst our sample was largely reflective of the UK student population in terms of demographics, our research is unable to comment on the impact of the course on minority groups that may be at particularly high risk of negative mental well-being [66]. Future research focusing on the impact of positive psychology courses on under-represented groups, and their experiences of well-being would be beneficial.

## Conclusions

We examined the impact of an online credit-bearing university positive psychology course on student well-being during the COVID-19 pandemic. Whereas students who completed the course showed stable levels of mental well-being and anxiety, students in a wait-list control group showed declines in the measures over time. Our findings suggest that positive psychology interventions are associated with a protective effect on student mental well-being in times of heightened environmental stress, such as during the COVID-19 pandemic. Higher education institutions would benefit from implementing positive psychology interventions into degree programmes to enhance student mental well-being.

## Supporting information

**S1 File. Full results for hypotheses one and two when adjusting for psychology degree programme.**
(DOCX)

**S1 Table. Mean change and standardised difference scores (Cohen's $d_z$) with 95% confidence intervals for SWEMWBS metric scores according to GRC response.**
(DOCX)

**S2 Table. Agreement with items relating to positive perceptions of university and interest in activities that can increase well-being in the intervention group at timepoint one.**
(DOCX)

## Acknowledgments

We would like to thank the participants and the mentors that led the happiness hubs.

## Author Contributions

**Conceptualization:** Sarah Jelbert, Laurie R. Santos, Bruce Hood.

**Data curation:** Catherine Hobbs, Sarah Jelbert, Bruce Hood.

**Formal analysis:** Catherine Hobbs.

**Funding acquisition:** Sarah Jelbert, Bruce Hood.

**Investigation:** Catherine Hobbs, Sarah Jelbert, Laurie R. Santos, Bruce Hood.

**Methodology:** Catherine Hobbs, Sarah Jelbert, Laurie R. Santos, Bruce Hood.

**Project administration:** Catherine Hobbs, Sarah Jelbert, Bruce Hood.

**Resources:** Catherine Hobbs, Sarah Jelbert, Bruce Hood.

**Supervision:** Sarah Jelbert, Laurie R. Santos, Bruce Hood.

**Visualization:** Catherine Hobbs, Sarah Jelbert.

**Writing – original draft:** Catherine Hobbs.

**Writing – review & editing:** Sarah Jelbert, Laurie R. Santos, Bruce Hood.

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
