## [Decision Letter · Decision Letter 0]

13 Jul 2021

PONE-D-21-17644

Evaluation of a credit-bearing online administered happiness course on undergraduates’ mental well-being during the COVID-19 pandemic

PLOS ONE

Dear Dr. Hobbs,

Thank you for submitting your manuscript to PLOS ONE. After careful consideration, we feel that it has merit but does not fully meet PLOS ONE’s publication criteria as it currently stands. Therefore, we invite you to submit a revised version of the manuscript that addresses the points raised during the review process.

We look forward to receiving your revised manuscript.

Kind regards,

Or Kan Soh

Academic Editor

PLOS ONE

Journal Requirements:

5. We note you have included a table to which you do not refer in the text of your manuscript. Please ensure that you refer to Table 2 and 3 in your text; if accepted, production will need this reference to link the reader to the Table.

Additional Editor Comments:

Dear Sir

Please make corrections to your manuscript based on the reviewers' comments.

Thank you.

Reviewers' comments:

Reviewer's Responses to Questions

**Comments to the Author**

1. Is the manuscript technically sound, and do the data support the conclusions?

Reviewer #1: Partly

Reviewer #2: Yes

2. Has the statistical analysis been performed appropriately and rigorously? 

Reviewer #1: Yes

Reviewer #2: Yes

3. Have the authors made all data underlying the findings in their manuscript fully available?

Reviewer #1: Yes

Reviewer #2: No

4. Is the manuscript presented in an intelligible fashion and written in standard English?

Reviewer #1: Yes

Reviewer #2: Yes

5. Review Comments to the Author

Reviewer #1: The authors studied UK undergraduate students during the COVID-19 pandemic and examined how well-being, anxiety, and perceptions of academic performance were affected by either participating in an 11-week positive psychology course in term 1 (intervention) or term 2 (control group). Measures were taken at three distinct timepoints, (T1) beginning of course, (T2) end of course, and (T3) 6 weeks after course completion. Differences in SWEMWBS and GAD-7 scores were observed across groups and timepoints (H1). In the intervention group they found that; SWEMWBS and GAD-7 scores remained relatively stable across the 3 timepoints, For the SHS measure the authors claim that they found “some evidence” that scores differed according to groups from timepoint 1 to 2 but “little evidence” of an interaction effect of group with timepoints 1 to 3 . In the wait listed control group, the researchers observed a decline in SWEMWBS scores, an increase in GAD-7 scores, while SHS scores fluctuated. They did not find any evidence that the course affected student perceptions of academic performance (H2). The authors concluded that enrollment in the course had a protective effect on psychological well-being and that the science of Happiness has beneficial effects on well-being regardless of the method of delivery (online vs in person).

This studies contribution to the core idea of a positive psychology course with lessons based around student engagement in PPI’s is interesting/promising. However, despite acknowledging that the random assignment of participants was not possible the authors continue to utilize causal language in their conclusions (e.g., 498-500, 514-516).

Below are some suggestions to strengthen the manuscript and clarify some key aspects of the method and the results.

1) • The authors should provide the readers with an explanation of the construct of Well-being in the introduction. The benefits of wellbeing on academic performance are discussed but not the construct itself. This is especially true as the distinction between happiness and wellbeing is mentioned later in the paper in interpreting the results. If there is consistency in positive psychology interventions in schools, it would be helpful to know which components are typical or essential. At my uni, there is a cottage industry of "happiness" courses arising from academic units including human development/family studies, English, and Philosophy, and it's unclear if those courses are as effective as courses on the psychological science of happiness.

2) It would help readers to have some additional context with regard to the features of the pandemic at play during the study. Were students residing on campus or had they gone home? Were they free to move or restricted? By whom and to what extent? Some previous research on the SARS pandemic, e.g. suggested that full government lockdowns were stressful. In this case, the extended uncertainty and political conflict over may have been more salient.

3) How much experience did the students have with online learning previously? If they were experienced online learners, they may have adapted differently than if they were moved into online learning with little choice or experience. Likewise, it would help to clarify how the class was delivered - other than the groups, was the class asynchronous? Or fully synchronous?

4) Related to to 2) and 3) In the abstract, it is implied that happiness courses are especially beneficial during stressful times. The data here do not appear to explicitly speak to that conclusion. Likewise, while the results worked online, the online environment during the pandemic is very different from well-planned, voluntary online education. It is more accurate to say the positive effects were seen in an online class delivered during a stressful time, compared to waitlist control.

5) The article is clear that the researchers did not assign students to classes, and that assignment was not random. Was it the case that those who were more engaged (responded first? followed up more?) were more likely to be placed in the class compared to waitlist? It's unclear how the psychology majors were so much more successful in getting in the class the first term.

6) The practical significance of the effects require more explanation for readers. The effects are quite modest. This does not mean they are not worth the cost, given the points the authors make about primary prevention and the infrastructure to reach up to all students in a challenging time. However, the effects that did reach statistical significance and reasonable statistical effect sizes remain quite small, clinically and functionally. For example, the difference in SWEMWBS well-being scores at time 2 and 3 was about 1 point. The authors of that scale suggest "If WEMWBS decreased by

three to eight points over the course of the project, WEMWBS would be demonstrating that

participant’s mental wellbeing meaningfully declined during the project" https://www.corc.uk.net/media/1244/wemwbs_practitioneruserguide.pdf. Thus, the students in WL control did not decline enough to be considered meaningful, generally. Likewise, participants in both the intervention and control groups had GAD-7 scores in the range of 7 - 9. According to the authors of the GAD-7, "Cut points of 5, 10, and 15 might be interpreted as representing mild, moderate, and severe levels of anxiety". That is all participants at all time points scored between mild and moderate anxiety, below the clinical cutoff for GAD. These are just very small, non-clinically significant differences among a number of non-significant differences the authors report. The small practical effects should be clearer to readers. This is not to say the results are not important to publish or the class is not worth the effort; only that clearer information is needed for researchers and practitioners in the future.

7) I appreciated the clear figures. I suspect the red/green colors without any other indicators (e.g. different symbols) will be challenging for colorblind readers to perceive.

8) Small thing: It was unclear in cases what was meant by a "unit". Is that the same as a "course"?

Reviewer #2: This paper presents a pre-registered study of the effectiveness of a 10 week Science of Happiness course on 1st year undergraduate students. If compared subjective wellbeing measures at baseline, end of the course and 6 weeks after the course and included a wait-list control group. The preregistration seems correct and consistent with the analyses conducted in the study. The study had good power and analyses were appropriate. The study found that the course may have been protective to mental health and that the effectiveness of this online course was similar to a previous in-person version of the same course. The introduction motivates the study well and references appropriate range of literature. The methods and results were accurate and mostly comprehensive. The discussion was measured and reflective of the data but could have included more speculation on future directions. I recommend acceptance with minor revisions.

Major questions & comments

1. Open Data: Could authors explain why an anonymised version of the dataset cannot be made available open access? It doesn't seem to me that this should be too difficult and would be a good improvement on the currently restricted data.

2. Figures 1 & 2 - would be better with more meaningful labels than timepoint 1,2,3 (e.g. Start of course, end of course, end + 6 weeks) For consistency I think you should also include the equivalent graph for SHS.

3. Methods - Need a bit more about the live sessions (Lines 165-169). Where these hosted by the lecturers. What kind of group activities were included? Mention how attendance was monitored.

4. Also need to be clearer about how the ratings were collected. Presumably this was with an online questionnaire tool of some kind? No details are given at present. (e.g. what platform? how were questionnaires delivered to participants, how long did they have to respond? were reminders sent. etc)

5. Statistical Analysis - The statistical analyses are well described but analysis code ought to be made open.

6. Discussion - Generally clear and consistent with results. Two things that didn't seem to be covered were 1. what mechanism is causing the programme to be effective? I don't think the study has data to address this but it would be beneficial for authors to speculate and provide hypotheses for future research. 2. It is unclear whether the same effect could be achieved with something less than a 10 week credit bearing program. Again I would welcome some speculation.

Minor

1. Corresponding author shown as Hobbs on PLOS system but as Hood on manscript.

2. I would mention preregistration in the abstract.

3. line 133 Participants - i would mention here how many were eligible to take part in intervention and wait-list and that, of these, how many were included in the study.

4. lines 291-293 - a bit more detail on how analyses were adjusted by degree programme.

5. line 383-385 - little evidence or no evidence?

6. lines 421-427 - provide N for this analysis.

7. lines 439 -433 - unwieldy sentence with long distance between subject and object. Break into two and . maybe rephrase to make it clearer that this is assessing relative level of engagement "However, level of engagement in the course did not influence change in well-being (SWEMWBS, etc). Nor was there any influence of perceptions of being at university ... "

8. line 686: Paper title is wrong.

6. PLOS authors have the option to publish the peer review history of their article (what does this mean?). If published, this will include your full peer review and any attached files.

Reviewer #1: No

Reviewer #2: **Yes: **Caspar Addyman

---

## [Author Response · Author response to Decision Letter 0]

8 Sep 2021

Thank you to the reviewers for their careful consideration of our manuscript and for their detailed comments, we have responded to each of their points below in bolded text.

Reviewer #1: 

This studies contribution to the core idea of a positive psychology course with lessons based around student engagement in PPI’s is interesting/promising. However, despite acknowledging that the random assignment of participants was not possible the authors continue to utilize causal language in their conclusions (e.g., 498-500, 514-516).

Response: We have amended our wording to be more speculative surrounding the causal nature of our effects (lines 555-556 & 571).

Below are some suggestions to strengthen the manuscript and clarify some key aspects of the method and the results.

1) • The authors should provide the readers with an explanation of the construct of Well-being in the introduction. The benefits of wellbeing on academic performance are discussed but not the construct itself. This is especially true as the distinction between happiness and wellbeing is mentioned later in the paper in interpreting the results. If there is consistency in positive psychology interventions in schools, it would be helpful to know which components are typical or essential. At my uni, there is a cottage industry of "happiness" courses arising from academic units including human development/family studies, English, and Philosophy, and it's unclear if those courses are as effective as courses on the psychological science of happiness.

Response: We have now included an explanation of the construct and importance of mental well-being in our introduction (lines 72-78). As this is a relatively new field there is currently no consistent approach to positive psychology courses in educational settings, and specific components and positive psychology interventions administered vary. We have expanded upon this in our discussion and suggest that future research evaluating the active components of the course would be beneficial (lines 634-650).

2) It would help readers to have some additional context with regard to the features of the pandemic at play during the study. Were students residing on campus or had they gone home? Were they free to move or restricted? By whom and to what extent? Some previous research on the SARS pandemic, e.g. suggested that full government lockdowns were stressful. In this case, the extended uncertainty and political conflict over may have been more salient.

Response: We have clarified the context of COVID-19 restrictions on the study sample in our introduction (lines 121-131). Students had the option to reside on campus or live at home, however we do not have data on living situation for the study sample. During the period of this study COVID-19 regulations fluctuated but included two national lockdowns and regional tier systems of which the site of this study was predominantly in the most restricted tier. Restrictions placed on the study sample were in line with COVID-19 regulations in England. Students were able to return home for Christmas within a specified period, but travel was restricted outside of this period. 

3) How much experience did the students have with online learning previously? If they were experienced online learners, they may have adapted differently than if they were moved into online learning with little choice or experience. Likewise, it would help to clarify how the class was delivered - other than the groups, was the class asynchronous? Or fully synchronous?

Response: We have now commented on previous experience of online learning in our introduction (lines 128-131). The study sample consisted of first-year undergraduate students. We can therefore speculate that they may have had some experience of online learning during the period in which secondary schools and colleges were closed from March to June 2020. However, as this consisted of approximately four months and extended closures were not anticipated, limiting the ability of schools to deliver full-time education remotely, we believe the study sample on average experience only limited online learning previously. We do not have data on participant’s previous experience of online learning that would allow us to comment further on this. However, we have no reason to believe that extent of online learning experience varied between groups, which could have influenced our findings. 

We have also clarified that participants in the intervention group attended live sessions simultaneously, whereas pre-recorded lectures and journal activities were completed on a weekly basis but at a time of participants’ choosing (lines 186-187, 196-198, 212).

4) Related to to 2) and 3) In the abstract, it is implied that happiness courses are especially beneficial during stressful times. The data here do not appear to explicitly speak to that conclusion. Likewise, while the results worked online, the online environment during the pandemic is very different from well-planned, voluntary online education. It is more accurate to say the positive effects were seen in an online class delivered during a stressful time, compared to waitlist control.

Response: We have amended the wording in our abstract to reflect on our findings in the specific context of the study (lines 25-27).

5) The article is clear that the researchers did not assign students to classes, and that assignment was not random. Was it the case that those who were more engaged (responded first? followed up more?) were more likely to be placed in the class compared to waitlist? It's unclear how the psychology majors were so much more successful in getting in the class the first term.

Response: Degree programmes differed as to which semester open units, such as the Science of Happiness, must be completed in. For the majority of degree programmes, students can only take an open unit in the second semester. However, psychology students are able to complete open units in either semester. To avoid a large imbalance in the number of students completing the course in each semester and make most effective use of teaching resources, administrative staff assigned the majority of psychology students to complete the course in TB1. At the point of enrolment on the course students were not aware of which semester they would be allocated to upon enrolment. We have clarified this in our manuscript (lines 168-171). 

6) The practical significance of the effects require more explanation for readers. The effects are quite modest. This does not mean they are not worth the cost, given the points the authors make about primary prevention and the infrastructure to reach up to all students in a challenging time. However, the effects that did reach statistical significance and reasonable statistical effect sizes remain quite small, clinically and functionally. For example, the difference in SWEMWBS well-being scores at time 2 and 3 was about 1 point. The authors of that scale suggest "If WEMWBS decreased by

three to eight points over the course of the project, WEMWBS would be demonstrating that

participant’s mental wellbeing meaningfully declined during the project" https://www.corc.uk.net/media/1244/wemwbs_practitioneruserguide.pdf. Thus, the students in WL control did not decline enough to be considered meaningful, generally. Likewise, participants in both the intervention and control groups had GAD-7 scores in the range of 7 - 9. According to the authors of the GAD-7, "Cut points of 5, 10, and 15 might be interpreted as representing mild, moderate, and severe levels of anxiety". That is all participants at all time points scored between mild and moderate anxiety, below the clinical cutoff for GAD. These are just very small, non-clinically significant differences among a number of non-significant differences the authors report. The small practical effects should be clearer to readers. This is not to say the results are not important to publish or the class is not worth the effort; only that clearer information is needed for researchers and practitioners in the future.

Response: We have now included a section in our discussion clarifying the magnitude of observed effects (lines 575-587). As the reviewer notes, although these were relatively small changes the Science of Happiness course and similar well-being courses are intended in part as preventative university-wide approaches to mental health. Whilst effects are therefore small at an individual level they are relevant when applied at a university campus level.

7) I appreciated the clear figures. I suspect the red/green colors without any other indicators (e.g. different symbols) will be challenging for colorblind readers to perceive.

Response: Thank you for highlighting this issue, we have amended the colour palette of our graphs to be perceptible to colour-blind readers.

8) Small thing: It was unclear in cases what was meant by a "unit". Is that the same as a "course"?

Responses: Optional academic courses completed by students that contribute to completion of their degree programme (i.e. provide a certain number of credits) are often referred to as open units. However, to avoid confusion we have amended our manuscript to refer to course rather than unit throughout.

 

Reviewer #2: 

1. Open Data: Could authors explain why an anonymised version of the dataset cannot be made available open access? It doesn't seem to me that this should be too difficult and would be a good improvement on the currently restricted data.

Response: Although we acknowledge that it would be preferable to publish the data as open access unfortunately it is not possible for this study. As we did not obtain explicit consent from participants to publish data as open access, we have been advised by the data repository service that it is only possible to publish that data as restricted access even if fully anonymised. It is possible to obtain the dataset using a data access agreement. We do not believe obtaining retrospective consent is feasible as anything other than a perfect response rate would result in the published dataset differing from that used in the analyses for this paper. We have amended our consent forms for future studies within this field to include open access publication to ensure all future data will be made open access.

2. Figures 1 & 2 - would be better with more meaningful labels than timepoint 1,2,3 (e.g. Start of course, end of course, end + 6 weeks) For consistency I think you should also include the equivalent graph for SHS.

Response: We have now included a figure for the SHS results (figure 3), and have amended our timepoint axis labels in line with the reviewers’ suggestion.

3. Methods - Need a bit more about the live sessions (Lines 165-169). Where these hosted by the lecturers. What kind of group activities were included? Mention how attendance was monitored.

Response: We have provided the additional details requested in our manuscript (lines 196-203). The live sessions were hosted by two of the co-authors of this study (SJ and BH), a lecturer and professor, respectively. The live session revisited the week’s pre-recorded lecture content, elaborating on examples and including activities such as pop quizzes, Q&As, replicating simple online experiments and group discussions. Individuals’ attendance at live sessions was not monitored, but overall attendance was typically around 80%. Recordings of the live session were also made available to students. We have clarified ‘monitored levels of engagement’ in lines 217 and 201-203.

4. Also need to be clearer about how the ratings were collected. Presumably this was with an online questionnaire tool of some kind? No details are given at present. (e.g. what platform? how were questionnaires delivered to participants, how long did they have to respond? were reminders sent. etc)

Response: We have included further details for our data collection method (lines 223-227). The online survey software Qualtrics was used to obtain questionnaire responses. Links to complete the survey were circulated to students using the courses’ online learning environment. Reminders to complete the survey were circulated approximately one week after the initial invite for each timepoint. Participants were required to respond within two weeks of the surveys being opened.

5. Statistical Analysis - The statistical analyses are well described but analysis code ought to be made open.

Response: We have now published our code as open access on Open Science Framework (https://osf.io/dvz3q/). The code is also available as part of the data repository.

6. Discussion - Generally clear and consistent with results. Two things that didn't seem to be covered were 1. what mechanism is causing the programme to be effective? I don't think the study has data to address this but it would be beneficial for authors to speculate and provide hypotheses for future research. 2. It is unclear whether the same effect could be achieved with something less than a 10 week credit bearing program. Again I would welcome some speculation.

Response: We have added a section to our discussion (lines 634-650) clarifying the need for further research on the active ingredients of the intervention and whether similar benefits may be obtained from an abbreviated version of the course. 

Minor

1. Corresponding author shown as Hobbs on PLOS system but as Hood on manscript.

Response: The PLOS ONE submission system automatically designates the submitting author as the corresponding author. It is not possible to edit this without restricting access to the manuscript. PLOS ONE advises that “Whoever is designated as a corresponding author on the title page of the manuscript file will be listed as such upon publication.” 

2. I would mention preregistration in the abstract.

Response: We now mention the pre-registration in the abstract (lines 18-19).

3. line 133 Participants - i would mention here how many were eligible to take part in intervention and wait-list and that, of these, how many were included in the study.

Response: We are not able to report a single number for students that were eligible to take part as this fluctuated based on students enrolling and disenrolling from the course. These numbers were particularly variable at the point of the baseline data collection with first year students frequently switching between open units in the first week of term.

4. lines 291-293 - a bit more detail on how analyses were adjust ed by degree programme.

Response: We have clarified that a binary variable indicating whether students were enrolled on a Psychology degree or another degree was included as an additional predictor in the regression models outlined for hypotheses one and two (lines 334-336).

5. line 383-385 - little evidence or no evidence?

Response: We have amended our wording to ‘no evidence’ (lines 429-431).

6. lines 421-427 - provide N for this analysis.

Response: We have reported the N for weekly happiness ratings (lines 462-463).

7. lines 439 -433 - unwieldy sentence with long distance between subject and object. Break into two and . maybe rephrase to make it clearer that this is assessing relative level of engagement "However, level of engagement in the course did not influence change in well-being (SWEMWBS, etc). Nor was there any influence of perceptions of being at university ... "

Response: We have amended the sentence structure in this section (lines 492-497).

8. line 686: Paper title is wrong.

Response: Thank you for identifying this, we have now corrected the paper’s title (lines 792-794).

---

## [Decision Letter · Decision Letter 1]

21 Jan 2022

Evaluation of a credit-bearing online administered happiness course on undergraduates’ mental well-being during the COVID-19 pandemic

PONE-D-21-17644R1

Dear Dr. Hobbs,

We’re pleased to inform you that your manuscript has been judged scientifically suitable for publication and will be formally accepted for publication once it meets all outstanding technical requirements.

Kind regards,

Stephan Doering, M.D.

Academic Editor

PLOS ONE

Reviewers' comments:

Reviewer's Responses to Questions

**Comments to the Author**

1. If the authors have adequately addressed your comments raised in a previous round of review and you feel that this manuscript is now acceptable for publication, you may indicate that here to bypass the “Comments to the Author” section, enter your conflict of interest statement in the “Confidential to Editor” section, and submit your "Accept" recommendation.

Reviewer #2: All comments have been addressed

2. Is the manuscript technically sound, and do the data support the conclusions?

Reviewer #2: Yes

3. Has the statistical analysis been performed appropriately and rigorously? 

Reviewer #2: Yes

4. Have the authors made all data underlying the findings in their manuscript fully available?

Reviewer #2: Yes

5. Is the manuscript presented in an intelligible fashion and written in standard English?

Reviewer #2: Yes

6. Review Comments to the Author

Reviewer #2: (No Response)

7. PLOS authors have the option to publish the peer review history of their article (what does this mean?). If published, this will include your full peer review and any attached files.

Reviewer #2: **Yes: **Caspar Addyman

---

## [Editor Report · Acceptance letter]

26 Jan 2022

PONE-D-21-17644R1 

Evaluation of a credit-bearing online administered happiness course on undergraduates’ mental well-being during the COVID-19 pandemic 

Dear Dr. Hood:

I'm pleased to inform you that your manuscript has been deemed suitable for publication in PLOS ONE. Congratulations! Your manuscript is now with our production department. 

Kind regards, 

on behalf of

Professor Stephan Doering 

Academic Editor

PLOS ONE